# Concepts of Nucleation in Polymer Crystallization

**Jun Xu** [1,*] , **Günter Reiter** [2,*] and **Rufina G. Alamo** [3,*]

[1] Advanced Materials Laboratory of Ministry of Education, Department of Chemical Engineering, Tsinghua University, Beijing 100084, China

[2] Institute of Physics and Freiburg Materials Research Center, Albert-Ludwig-University of Freiburg, 79104 Freiburg, Germany

[3] FAMU-FSU College of Engineering, Department of Chemical and Biomedical Engineering, 2525 Pottsdamer St, Tallahassee, FL 32310, USA

* Correspondence: jun-xu@mail.tsinghua.edu.cn (J.X.); guenter.reiter@physik.uni-freiburg.de (G.R.); alamo@eng.famu.fsu.edu (R.G.A.)

**Abstract:** Nucleation plays a vital role in polymer crystallization, in which chain connectivity and thus the multiple length and time scales make crystal nucleation of polymer chains an interesting but complex subject. Though the topic has been intensively studied in the past decades, there are still many open questions to answer. The final properties of semicrystalline polymer materials are affected by all of the following: the starting melt, paths of nucleation, organization of lamellar crystals and evolution of the final crystalline structures. In this viewpoint, we attempt to discuss some of the remaining open questions and corresponding concepts: non-equilibrated polymers, self-induced nucleation, microscopic kinetics of different processes, metastability of polymer lamellar crystals, hierarchical order and cooperativity involved in nucleation, etc. Addressing these open questions through a combination of novel concepts, new theories and advanced approaches provides a deeper understanding of the multifaceted process of crystal nucleation of polymers.

**Keywords:** nucleation; polymer crystallization; primary nucleation; secondary nucleation; non-equilibrium





## 1. What Makes Nucleation of Polymer Crystals so Difficult?

The classical path for crystallization of any substance is via nucleation and growth. The crystallization of polymers passes along the same route; an initial nucleation step (primary nucleation) is followed by a growth process. However, as shown by numerous experiments for many polymers, the connectivity of a large number of monomers in these chain-like molecules makes homogeneous nucleation to be an extremely slow process, often leading to highly unstable systems of low crystallinity prone to age and change in time. To eliminate this shortcoming of polymers, several approaches are available for enhancing the nucleation probability. For example, the use of nucleating agents [1], self-seeding strategies [2,3], epitaxy [4,5], or the application of external forces (shearing) [6,7] have dramatic effects on the nucleation probability. Sometimes, blending with amorphous (incompatible) components may improve the nucleation rate [8]. Alternatively, various thermal protocols in processing polymer systems had been established, which allowed partially or fully circumventing the nucleation step [9–13], some representing unique possibilities for crystallizing polymers.

In many cases, well-established procedures used extensively for the crystallization of small molecules have been adapted and implemented for crystallizing polymer systems. However, there are also some features, which rely on the chain-like nature of polymers and thus cannot be found for systems of small molecules. Here, we would like to shed some light on already identified and potentially existing differences between small molecules and polymers with respect to their impact on nucleation mechanisms. We present and discuss a variety of mechanisms, which can initiate a polymer crystal: primary vs. secondary

nucleation, self-nucleation vs. self-induced nucleation, self-seeding . . . We discuss how chain conformations and a change in topology in the amorphous melt may affect the mechanisms and the kinetics of nucleation of polymer crystals.

Our views are only tentative and incomplete and thus invite amendments and complementary contributions. The presented ideas are often speculative or debatable. Thus, we anticipate that our text will provoke commentaries or criticisms, which are highly welcome. A frank but respectful discussion of open questions in the field of polymer crystallization will help develop new ideas and foster new concepts.

## 2. Beyond Thermodynamic Concepts

Typically, most concepts of nucleation employ thermodynamic parameters to express the change in free energy between the melt and the developing nuclei. Thermodynamic principles may be justified if characteristic times for establishing (local) equilibrium are much shorter than the characteristic time of nucleation. This imposes that the process of nucleation is slower than the time required for identifying and establishing the state of the lowest free energy. At the small length-scales of the size of a nucleus, we may assume that a polymer system is at equilibrium, implying that nucleation starts from a locally equilibrated polymer melt (equilibrium conformations). Often, the final crystalline state is characterized by a stem length or a number of folds, which are also treated as a local equilibrium state having minimum free energy within this local space. However, the resulting lamellar crystals are metastable and thus may change in time with often extremely slow characteristic relaxation processes for long polymers [14].

The formation of a nucleus for a polymer crystal requires conformational changes of the involved chains. Especially when chains interweave with others (entangled melts), equilibration processes can be very slow, and the time required for finding the state of lowest free energy for an ensemble of chains becomes increasingly long. Examples of such states have been found for melts of random copolymers [10,15]. For a broad spectrum of relaxation processes, polymer nucleation may become a multi-stage process. It is not obvious if conformational changes required for the formation of a (homogeneous) nucleus are the same when attaching polymers at the front of a growing crystal. Thus, differences may exist between primary nucleation and secondary nucleation.

Accordingly, we raise the following questions:

1. Do the characteristic parameters of a nucleus (e.g., shape, stem length, number of stems or chains involved . . . ) depend on how fast the nucleus was established? For a given temperature, is it possible to define (and identify) an "equilibrated" melt state? Do we obtain different types of nuclei if polymer melts are not equilibrated? For example, is there a difference between a nucleus formed in an equilibrated melt and a nucleus formed in a non-equilibrated (e.g., sheared) melt? In this context, the observation of a "melt memory" suggests that thermal history and melt annealing time can affect characteristic parameters of a nucleus (size, shape, number of stems . . . ). Both have been found relevant, affecting the early-stages of crystallization and hence, primary nucleation [10,15,16].

2. Connectivity of the monomers represents a characteristic feature of polymer chains. Not all monomers of any long-chain (having a large number of connected monomers) are involved in the formation of a nucleus. Accordingly, if some monomers of a chain are embedded in a nucleus, what happens to the others? Is the probability of nucleation affected by the existence of a first nucleus? Is the "nucleus environment" propagating along the backbone of the polymer chains? Can the remaining molten segments form another nucleus aided by such propagation?

3. If monomers from several chains are integrated into a nucleus, can that lead to an enhanced probability of nucleation along the backbone of such a correlated chain ensemble? Can we establish a relation to a process, which may be called "self-induced nucleation," on a fold surface [17–22]? Is it possible to draw an analogy to a bouquet of flowers, where a correlation exists between the stems (arranged in an orderly fashion) and the rather randomly arranged blooms (see Figure 1)?

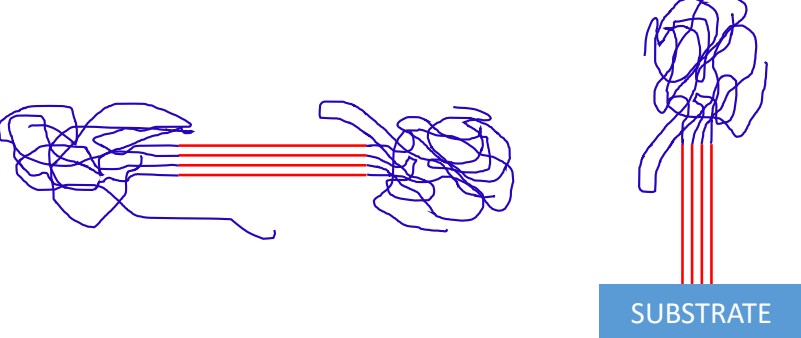

**Figure 1.** Schemes for a "bunch of flowers" of several polymer chains, which are partially involved in the formation of a nucleus (correlated stems) and partially staying amorphous ("bloom").

Is such a coupling between crystalline and amorphous segments capable of enhancing the nucleation probability? Can the nucleation probability of (non-equilibrated) polymer chains be derived from thermodynamic arguments, or is the nucleation probability rather of a kinetic nature with rates proportional to the number of chains correlated in the "bunch of flowers"? Can we draw an analogy to sheared polymer solutions, where stretched polymer chains tend to aggregate, leading to a process similar to phase separation? (See the definition of free energy of a system of stretched chains within a mean-field approximation given by Subbotin and Semenov [23]).

## 3. Nucleation from Non-Equilibrated Melts

When heating polymer crystals to a temperature above their observed melting point, often the crystallographic registry is lost, but the chain segments participating in the crystallites remain in close proximity, retaining some of the initial orientation due to a relatively slow thermal mobility. Recrystallization from such non-equilibrated melts is enhanced as the localized regions of lower entropy confer a self-nucleating melt structure. For most homopolymers, the enhanced recrystallization is limited to just 2–5 degrees above the observed melting. Crystalline melt-memory is not observed in homopolymers above their equilibrium melting temperature, and such incomplete sequence randomization in the melt disappears after increasing holding time leading to reproducible, equivalent crystallization kinetics [24–29]. Most experimental observations related to crystalline melt-memory in homopolymers can be explained on the grounds of thermodynamic phase behavior. Below the equilibrium melting temperature, the polymer melt is undercooled, and, besides the lack of a fast sequence diffusion and homogenization, the possibility of a small fraction of crystallites surviving in the melt cannot be excluded. In some examples, over 250 min were needed to erase self-nuclei [29,30].

What raises questions about the nature of such non-equilibrated melts are recent works in random ethylene 1-alkene copolymers that show memory of crystallization even at temperatures ca 30 degrees above their equilibrium melting point (~65 degrees above the observed final melting) [10–12,15,16]. This unusually strong melt-memory of copolymers is in sharp contrast with the behavior of linear polyethylene fractions and was associated with the process of sequence partitioning during the copolymer's crystallization [10]. As shown schematically in Figure 2, due to the branches being rejected from the crystalline regions, the path of selecting and dragging ethylene sequences to build copolymer crystallites generates a complex topology of branches, knots, loops, ties and other entanglements in the intercrystalline regions, especially at high levels of transformation [10,31]. When the crystallites of ethylene copolymers melt, clusters from the initial crystalline ethylene sequences remain in close proximity even at very high temperatures because segmental melt diffusion to randomize all sequences is hampered by branches and the constrained intercrystalline topology (Figure 2b). Melts with this type of memory are metastable systems where the clusters are effective "*pre-nuclei*" that only fully dissolve into a homogeneous melt at temperatures well above the equilibrium

melting point (Figure 2c) [10–12,15,16,32]. Segmented thermoplastic polyurethanes are also examples of systems with selective sequence crystallization and have also shown relatively strong melt memory [33].

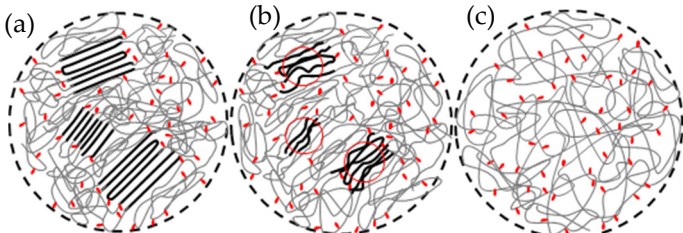

**Figure 2.** Schematics of (**a**) random copolymer semicrystalline structure with co-units (red dots) excluded from the crystalline regions; (**b**) heterogeneous, non-equilibrated melt with crystalline memory: (**c**) homogeneous equilibrated melt. Adapted with permission from Reference [10]. Copyright (2013) American Chemical Society.

The experimental evidence is consistent with the kinetic nature of melt memory [10–12, 15,16,24–26,34–37]. Even when cooling from the same melting temperature, the increase of crystallization temperature depends on molecular weight, the initial level of crystallinity or on how the standard crystalline state is prepared. Further studies of the effect of annealing time and molecular weight on the strong crystalline melt-memory of copolymers indicated that dissolution of such clusters, albeit thermally activated, is a very slow process. The copolymer's melt memory persists even after > 1000 min annealing, which is unexpected on the basis of prior self-diffusion and melts relaxation times for the same copolymers [15]. The very long characteristic times associated with the dissolution of melt memory observed for HPBDs contrast with relaxation times extracted from various avenues for the same systems [15]. For example, prior work on melt diffusivity for HPBDs at temperatures between 140 and 180 °C with a molecular weight ranging from 10.000 to 500.000 gmol$^{-1}$ estimated single-chain relaxation times between 1 ms and 10 s [38–41]. These values are 3–7 orders of magnitude smaller than the characteristic time for dissolution of memory. Hence, dissolution of melt memory appears to entail more than just reptation of polymer chains because the process of diffusing and randomizing ethylene segments from the initial crystallites is a lot costlier than classical translational chain diffusion. Such a discrepancy points to a dissolution process of memory that is not correlated with the single-chain dynamics.

What is the topology of the chain segments that emanate from the core crystalline lamellae that make the dissolution of memory (pre-nuclei) such a slow process? This feature must be explained beyond thermodynamic concepts, as mentioned earlier. Could such non-equilibrated melts be considered as metastable states that may have originated during crystallization as posited? [42] While testing the latter may be feasible with reliable isothermal experimental data, the small mass fraction involved in efficient self-nuclei in melts with memory may hamper direct experimental detection.

## 4. Nucleation on the Fold Surface of Polymer Lamellar Single Crystals

In three-dimensional crystalline polymer samples (bulk samples), experimental observations often reveal stacks of a large number of intimately linked lamellae, ideally all of them parallel to each other [43,44]. Even in spherulites, parallel lamellae, rather than randomly oriented ones, represent the rule [44]. If it is assumed that each lamella nucleates independently, the result would be an array of randomly oriented, quasi-two-dimensional lamellar crystals statistically distributed in space. These findings suggest that a fundamental mechanism exists, which allows orienting and aligning many lamellae parallel to each other. However, amorphous layers represent a barrier to crystal growth. Thus, special nucleation mechanisms are required for lamellar single crystals to initiate the formation of crystalline layers atop an amorphous fold surface.

Furthermore, experiments have clearly shown that such stacks can be initiated (nucleated) off-center and multiple times on a basal lamellar single crystal [20,21]. Most surprisingly, all stacks of lamellae found atop one basal lamella exhibited a unique orientation, identical to the orientation of the underlying single crystal.

Even more surprisingly, there is evidence that all lamellae in such a stack are in the crystallographic registry. Scattering patterns from such three-dimensional crystalline superstructures have signatures of single crystals. Obviously, an initially formed (basal) crystalline mono-lamella can transfer information on crystal orientation to additional (multiple) lamellar layers forming atop its fold surface, which is a special case of surface-induced nucleation.

However, the exact mechanism of producing multilayer stacks of lamellae and transferring crystal orientation is not obvious. Different mechanisms have been considered, such as macro-screw dislocations [45–47], lamellar branching due to dangling chain [48], or the insertion of polymer chains into the gap between two branches of a dendritic lamellar crystal [22].

Already in 1972, in the context of the "self-decoration" of polymer single crystals, Kovacs and Gonthier [17] have identified different types of decorating sites, which can be encountered on poly(ethylene oxide) (PEO) single crystals grown from the melt (see Figure 3). This includes sharp edges of basal lamellae (eb) or spiral terraces (es), double layers (ed) or large thickened portions (ee), all involving row nucleation. Surface decoration involving needle-like nucleating sites may arise either from rather large protrusions (p) or smaller irregularities (i), which may uniformly cover the entire surface of the crystals.

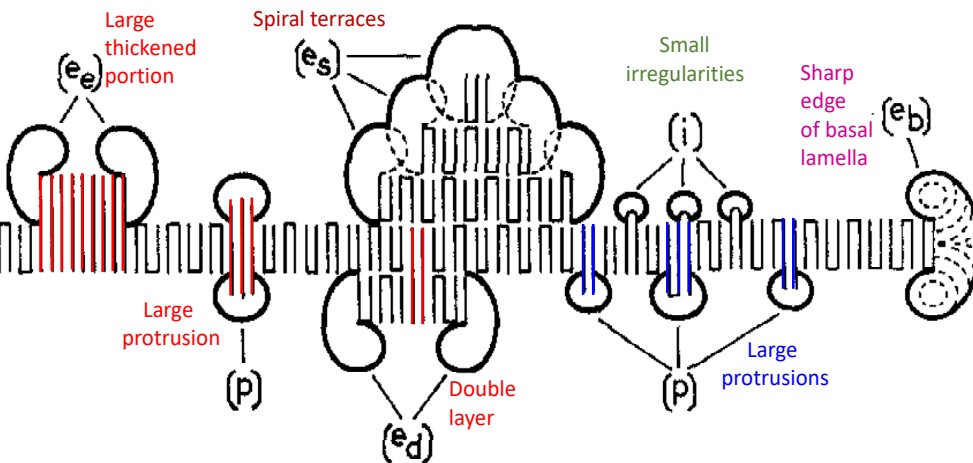

**Figure 3.** Schematic cross-section of a self-decorated poly(ethylene oxide) (PEO) crystal lamella, showing the various decorating sites, which can be encountered on PEO single crystals grown from the melt (note that the thickness of the basal lamella is about 100 times enlarged with respect to the size of the decorating units). Adapted by permission from ref. [17] Copyright 1972, Springer.

Kovacs and Gonthier assumed that such decoration/nucleation sites might cover the entire surface of lamellar crystals uniformly. The appearance of such sites was attributed to a statistical process, leading to randomly distributed sites.

Even at present, it is not yet known how the different layers in stacks of correlated lamellar crystals are connected to each other and what properties the interfacial layers have. It is expected that lamellae are connected by "tie chains". However, it may be debated if there exist amorphous sequences between the crystalline stems or if, as suggested by Kovacs and Gonthier, the link is established by an all-crystalline sequence. Extensive experimental evidence from the past is conclusive that the extent and topology of the interlamellar region depend on molecular weight and crystallization mode [43,49]. It is also unknown if and how properties of the interfacial amorphous layers sandwiched between crystalline lamellar cores are affected by the way lamellae are linked via tie chains. On one

hand, tie chains may help to distribute the mechanical load between neighboring lamellar layers, beneficial for improving mechanical properties. On the other hand, the existence of tie chains may impose constraints on the mobility of chain segments of the amorphous interlayers. Such tie chains may also be involved in the imperatively required nucleation process for initiating each and every crystalline lamellae within such stacks of correlated lamellae, similar to the suggestions of Kovacs and Gonthier.

Using atomic force microscopy (AFM), it could be shown that all lamellae in such a stack had the same thickness, even when the stack was clearly nucleated off-center of the basal lamella [20–22]. All secondary lamellae showed the same orientation as the basal lamella. The nucleation density $n_S$ was defined as the number of secondary lamellae observed per unit area on the amorphous fold surface of a lamellar crystal [20]. $n_S$ was found to be many orders of magnitude higher than homogeneous nucleation in the surrounding thin-film. In addition, a significant dependence of $n_S$ on both crystallization temperature and the initial film, the thickness was observed. The probability of generating correlated lamellae was found to be controlled by lamellae branching orthogonally from a single primary (basal) lamella. $n_S$ was related to the width $w$ of the branches of the primary lamella such that $n_S \sim w^{-2}$. This relation was independent of molecular weight, crystallization temperature, and film thickness.

A nucleation mechanism, termed "self-induced nucleation", based on the insertion of polymers into a branched primary lamellar crystal, has been proposed (see Figure 4) [22]. The space in between crystalline lamellae branches is identified as sites for nucleation on the amorphous fold surface. We note that central aspects of the proposed mechanism are similar to nucleation of "giant screw dislocations" induced by "lacunae" (representing "*small, almost imperceptible 'slits' that may develop naturally in the growth process*") identified by Keith and Chen [18].

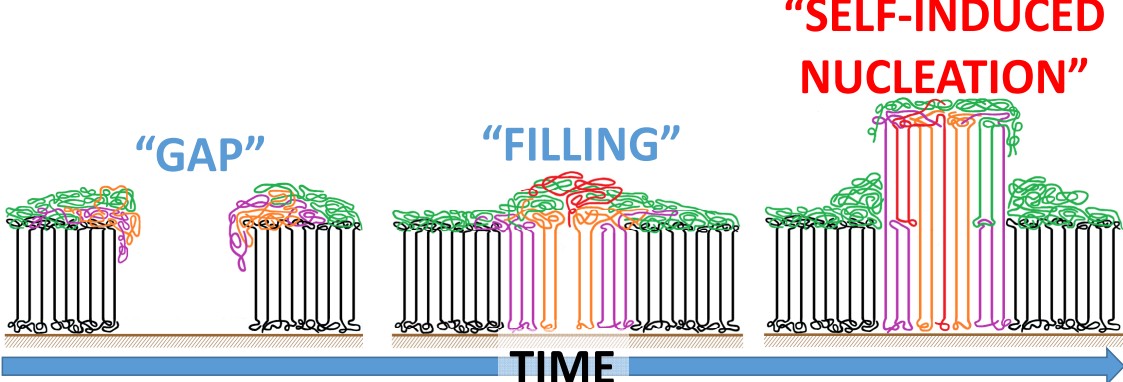

**Figure 4.** Scheme demonstrating self-induced nucleation, a new mechanism for the formation of secondary lamellar crystals, insertion of polymer chains in the impinging corner of two primary branches. Adapted from ref [22].

Based on inserted polymers, two superposed lamellae share polymer chains, which have the potential to have identically oriented crystalline monomers segments, bringing neighboring lamellae to the registry.

It is highly important that confirmed by results from several groups, stacks of lamellar crystals were also observed for diblock copolymers. Thus, even a thick amorphous interlayer does not inhibit the nucleation of stacks of correlated crystalline lamellae. However, details of this nucleation mechanism are still a matter of debate. Understanding the underlying physics of this nucleation mechanism allows tuning the areal density of tie chains within stacks of highly correlated polymer single crystals.

## 5. Secondary Nucleation on the Lateral Growth Front of Polymer Lamellar Crystals

The temperature coefficient of the selection of the thickness and the radial growth rate of polymer lamellar crystals are typically related to secondary nucleation, a process

occurring repetitively on the growing crystal surface. However, detailed information on secondary nucleation on a molecular level is not yet completely available. Many questions remain still open: Do polymers order exclusively only when they are already in contact with the growth surface (one-stage), or do polymers start to preorder in the melt close to the growth surface and eventually merge with ordered polymers previously integrated at the growth front (two or multiple stages)? Does secondary nucleation take place by attaching stems (ordered sequences of monomers) of a defined size (length) one-by-one, or is such nucleation requiring the concurrent thickening and widening of a cluster of simultaneously attached stems? Which parameters determine the resulting lamellar thickness and the way chains get folded (e.g., the number of adjacently reentering folds) during crystal growth?

Most of these questions are not addressed by the various theories developed for polymer crystallization. For example, approaches proposed by Lauritzen and Hoffman [50, 51], Sadler and Gilmer [52,53], or the intramolecular nucleation theory [54–56] and the continuum crystallization theory [57] assume that polymer chains order in a single-stage deposition process at the growth front. By contrast, Strobl proposed that polymer chains undergo a multi-stage ordering process, where the molten chains first transform into a mesophase before they finally become part of the resulting crystal [58,59].

Theories and models of one-stage crystal growth differ with respect to the details of molecular processes. According to the Lauritzen–Hoffman theory (LH), the first adsorbed stem is considered as the critical secondary nucleus. The length of this stem is expected to be constant in the course of crystal growth but depends on temperature. After forming the critical nucleus, LH expects that the chain folds and consecutively attaches further stems at the growth front. Later developed approaches considered fluctuation of the length of the attached stem. In the Sadler–Gilmer model, chains can attach and fold at any stem length. The resultant (mean) lamellar thickness reflects the kinetics of attaching and detaching these stems of different lengths. In the intramolecular secondary nucleation, the whole chain or a part of one chain (depending on the molecular weight) forms a two-dimensional secondary nucleus, which consists of multiple stems from the same chain.

In classical nucleation theories with capillarity approximation (accounting for interfaces), the change in bulk free energy per volume and the corresponding change in interfacial free energy per area does not vary with nucleus size. For a two-dimensional secondary nucleus, the free energy change is:

$$\Delta G = -bl_c w \Delta g + 2bl_c \sigma + 2bw\sigma_e \tag{1}$$

where $l_c$ and $w$ is the thickness and width of the two-dimensional secondary nucleus, respectively. $b$ is the width of a layer of stems and $\sigma_e$ and $\sigma$ are the basal and lateral surface free energies, respectively. With the fixed volume, $bl_c w$ of a secondary nucleus, Equation (1) gets a minimum when:

$$\frac{l_c}{w} = \frac{\sigma_e}{\sigma} \tag{2}$$

A two-dimensional nucleus with the shape according to Equation (2) reaches the maximum of $\Delta G$ at the critical lamellar thickness:

$$l_c^* = \frac{2\sigma_e}{\Delta g} \tag{3}$$

where $\Delta g$ stands for the bulk free energy change per volume, respectively. When the lateral dimensions are infinitely large, the critical thickness given by Equation (3) is also the minimum stable thickness. However, for a two-dimensional secondary nucleus, this width is finite and rather comparable to the length of the stem.

Given the fact that a critical two-dimensional nucleus is unstable [60], stability can be achieved, for example, by *doubling* the thickness of the critical nucleus [60,61]. The

minimum lamellar thickness, $l_{c,min}$, for the formation of the smallest stable two-dimensional nucleus ($\Delta G = 0$) can be obtained by Equation (4) [61], which differs from $l_c^*$.

$$l_{c,min} = \frac{4\sigma_e}{\Delta g}, \qquad w_{min} = \frac{4\sigma}{\Delta g} \qquad (4)$$

In fact, Equation (4) is identical to the critical thickness for a 3D nucleus, which does not need to thicken to become stable.

Based on the stochastic process of nucleation in melts of random copolymers and polymer blends, we have recently examined the size of critical secondary nuclei [62,63] (Figure 5). Within the studied temperature range, each secondary nucleus of poly(butylene succinate) consists of several stems. The number increases with crystallization temperature from 5 to 8 stems. At low temperatures, these stems are contributed by (on average) 2 different chains (reflecting a combination of intra- and inter-chain secondary nucleation), while at higher temperatures, a single chain is sufficient for the formation of the "multi-stem nucleus". We note that the classical Lauritzen–Hoffman theory assumed that a single stem could act as a critical secondary nucleus at the crystal growth surface.

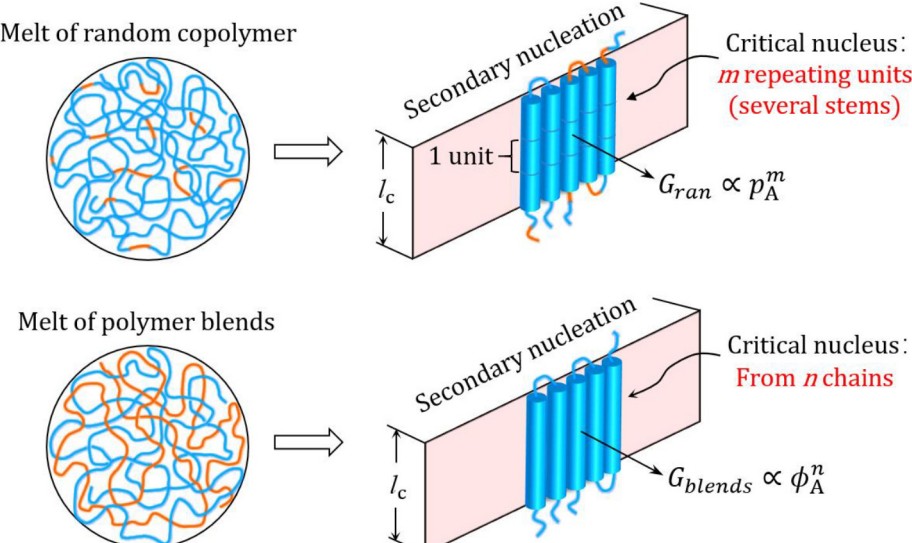

**Figure 5.** Scheme for determining the size of a critical secondary nucleus based on the crystallization kinetics of random copolymer and miscible crystallizable/amorphous polymer blends. Adapted with permission from Reference [62]. Copyright 2019, American Chemical Society.

Based on our observations, we suggest that the size of a critical secondary nucleus typically involves more than one stem and often more than one polymer chain. Further studies, e.g., via computer simulation, may unveil the fundamental molecular processes, in particular their kinetics, involved in secondary nucleation at the growth front of lamellar polymer crystals.

## 6. Kinetics of Molecular Processes Involved in Nucleation of Crystals

Due to the connectivity of monomers within a polymer chain, the kinetics of nucleation (both primary and secondary) of polymer lamellar crystals differs from that of small molecules. Classical nucleation theories for small molecules originated from the work of Volmer and Weber [64], Becker and Döring [65], and Turnbull and Fisher [66]. Assuming that diffusion is not the rate-limiting step and based on a Markovian process for attaching and detaching one unit at a time, the kinetics of nucleation is mainly determined by the formation of critical nuclei that must overcome a high free energy barrier. In this Markov process of attaching and detaching small molecules from a nucleus of a molecular crystal,

the attachment probability per surface area is assumed to be constant, i.e., independent of the size of the nucleus [66] (see Figure 6).

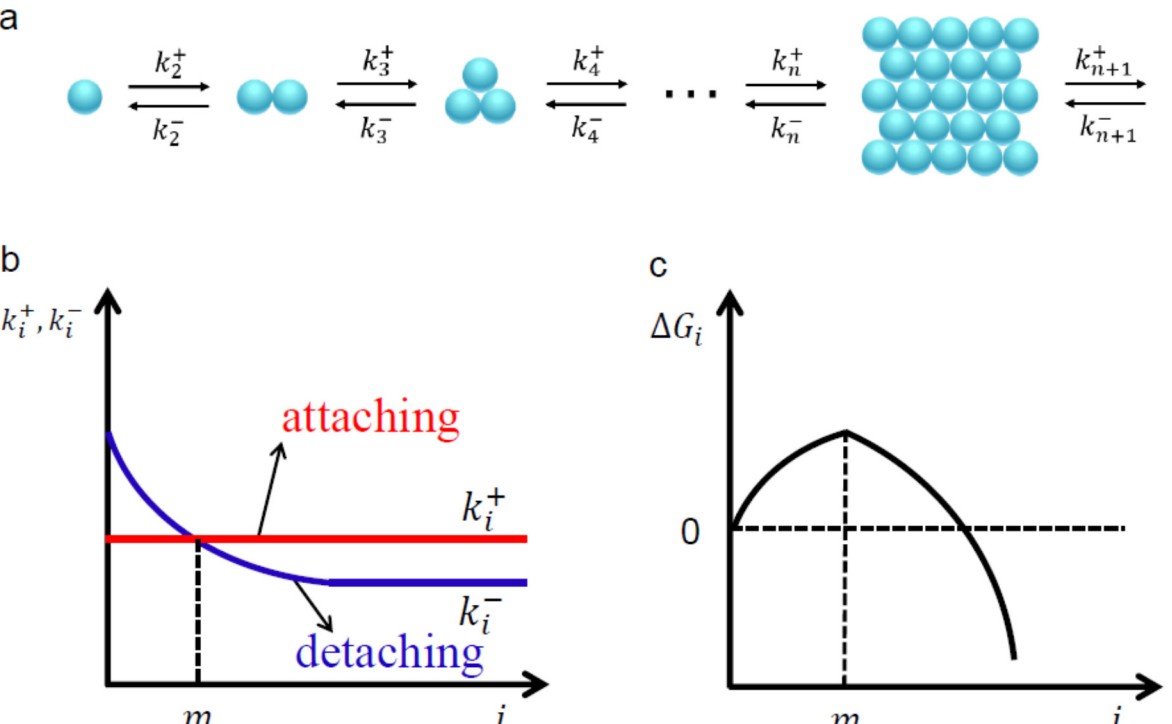

**Figure 6.** Microscopic kinetics of nucleation of small molecular crystals. (**a**) Scheme showing the microscopic process of attaching and detaching. $k_i^+$ and $k_i^-$ indicate the rate constant for attaching a molecule to form a nucleus containing $i$ molecules and detaching a molecule from a nucleus containing $i$ molecules; (**b**) variation of the rate constants of attaching and detaching with nuclei size; (**c**) free energy change of formation of nuclei containing different numbers of molecules. The symbols $i$ and $m$ indicate the number of molecules within the nucleus and that within the critical nucleus, respectively.

As a result, the nucleation rate $J$ can be obtained by assuming a transient steady state:

$$J \sim \exp\left(-\frac{E + \Delta G^*}{k_B T}\right) \tag{5}$$

where E and $\Delta G^*$ indicate the energy barrier for a unit to diffuse across the melt/crystal interface and the barrier to form a critical nucleus, respectively.

Strictly speaking, for a cascaded process consisting of multiple steps for each attachment/detachment, "equilibrium" means that the net flux at each step $J_i$ is zero (i.e., the difference in chemical potential is zero):

$$J_i = k_i^+ C_{i-1} - k_i^- C_i = 0 \tag{6}$$

$$\frac{C_i^{eq}}{C_{i-1}^{eq}} = \frac{k_i^+}{k_i^-} \tag{7}$$

$$\frac{C_i^{eq}}{C_1^{eq}} = \prod_{j=2}^{i} \frac{k_j^+}{k_j^-} \tag{8}$$

where $k_i^+$ and $k_i^-$ represent the rate constants for attaching a molecule leading to a nucleus with $i$ molecules and detaching a molecule from a nucleus with $i$ molecules. $C_i^{eq}$ is the



equilibrium concentration of nuclei with each containing $i$ molecules. In the nucleation stage, the chemical potential can be calculated from the microscopic kinetics:

$$\Delta g_i = \Delta G_i - \Delta G_{i-1} = k_B T \ln \frac{k_i^-}{k_i^+} \tag{9}$$

$$\Delta G_i = \sum_{j=2}^{i} \Delta g_j = -k_B T \ln \left( \prod_{j=2}^{i} \frac{k_j^+}{k_j^-} \right) \tag{10}$$

where $\Delta G_i$ is the total free energy change involved in forming a nucleus consisting of $i$ molecules. $\Delta g_i$ is the free energy change after attaching a molecule from the amorphous state to form a nucleus with $i$ molecules. $k_B$ and $T$ indicate the Boltzmann constant and the absolute temperature, respectively. Combining Equations (8) and (10), we have:

$$\frac{C_i^{eq}}{C_1^{eq}} = \exp\left(-\frac{\Delta G_i}{k_B T}\right) \tag{11}$$

The above equations show that the concentration of each size of nuclei equals its equilibrium value. This can only be achieved when the flux at each step is zero, i.e., no crystallization could be observed. For example, this is the case in the solution below the supersaturation point or in the melt without supercooling.

During crystallization, due to supersaturation or supercooling, there is a positive flux for each step; namely, the concentration of nuclei is below the equilibrium value enabling crystallization at a certain net rate until most of the material has been crystallized. For a crystal with a size larger than the minimum stable size, the equilibrium concentration should be larger than the bulk value. Since the cluster concentration could not be larger than the bulk value, the system during crystallization is out of equilibrium. However, if the crystallization rate is small compared to the rate of attaching and detaching, the system can be considered near-to-equilibrium. Without considering the diffusion effect, larger supersaturation or higher supercooling would lead to a lower concentration of the nuclei compared to its equilibrium concentration, thus a higher nucleation rate.

### 7. Differences in the Kinetics of Nucleation for Polymer Lamellar Crystals and for Small Molecular Crystals

The assumption of a Markovian process (i.e., the kinetics of attaching/detaching a molecule to/from a crystal surface is independent of the presence of other molecules) in small molecular crystals is not valid for polymer crystallization. For intra-chain nucleation of polymer lamellar crystals from the melt, the probability of attaching an additional unit from the same polymer chain requires that this unit has the right conformation and thus decreases exponentially with the number of units from this chain [52,53] (as shown in Figure 7), which are already adsorbed on the crystal surface. Accordingly, attaching units on a polymer chain is a non-Markovian process, and nucleation theories developed for the crystallization of small molecules cannot be simply translated to intra-chain nucleation of polymers.

The influence of connectivity is clearly revealed by the kinetics of how polymer crystals grow in size in the direction along the chain axis. For crystals of small molecules, their size increases linearly with growth time. However, the lamellar thickness of a polymer crystal increases only proportional to the logarithmic of the growth time [14,44,45], a consequence arising from the decreasing probability to attach additional units from the same polymer chain in the direction of the lamellar thickness.

In the Markovian process relevant for small molecules, the attachment probability and the resulting growth kinetics are not affected by previously attached molecules. On the contrary, in the non-Markov process of attaching monomers from the same polymer chain, the growth kinetics is affected by previously attached monomers of this chain, reflecting an

entropic barrier, which attaching neighboring units from the same chain must overcome when choosing a suitable (nearby) attachment site [52,53].

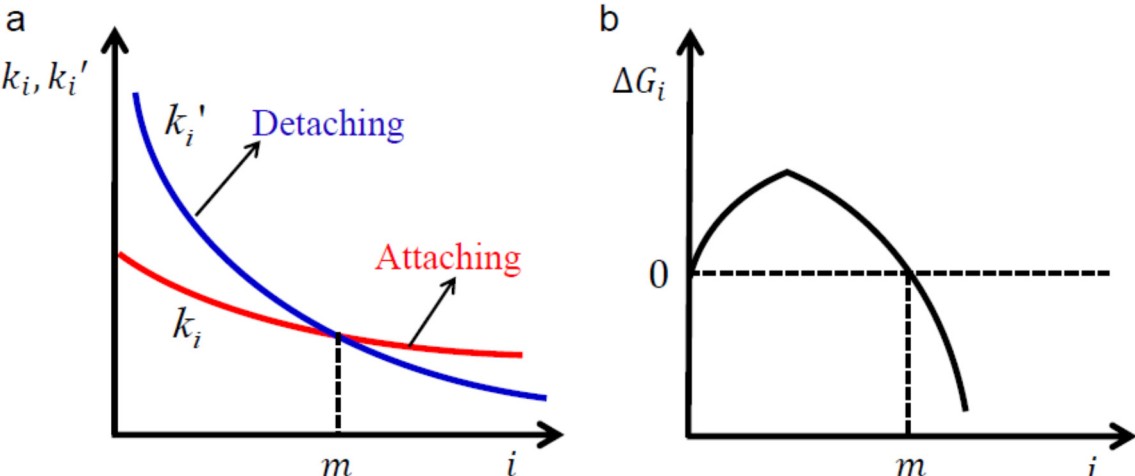

**Figure 7.** Microscopic kinetics of attaching and detaching for flexible polymer chains. (**a**) Variation of the rate constants of attaching and detaching with a number of monomers; (**b**) free energy change of formation of nuclei containing a different number of monomers. The symbol m indicates the number of monomers within the minimum stable nuclei for crystals of flexible polymers.

Therefore, we must determine when this entropic barrier becomes negligible. Theoretically, the attachment probability of monomers will be independent of other (already attached) monomers if monomers are sufficiently far apart from already attached ones or if they belong to different chains. Consequently, for intra-chain nucleation, due to the covalent linking of all monomers within a polymer chain, the kinetics of nucleation should be governed by an additional entropic barrier due to chain connectivity.

To derive the rate of secondary nucleation, Lauritzen and Hoffman [67] assumed that the attaching/detaching unit is a stem (a sequence of a certain number of monomers) and that attachment/detachment can be described by a Markovian process like for nucleation of crystals of small molecules. The assumption of a Markovian process implies that stems are independent and already attached stems do not affect the attachment of other stems, which, therefore, may also be chosen from other chains: Such independence of stems cannot explain the often-observed adjacent reentry of folded chains in polymer lamellar crystals.

Accounting for a non-Markovian process of intra-chain nucleation, it is proposed that the nucleation barrier of polymer lamellar crystals should be determined by the product of $l_{c,min}$ and the number of stems within a nucleus rather than the previously considered $l_c^*$, introducing an entropic barrier for the increase of the lamellar thickness.

## 8. Metastability of Polymer Crystals

The kinetics of secondary nucleation, i.e., the fastest net rate of the structure formation, determines the path and the rate of polymer crystallization and thus the final structures. The resulting structures, which are not representing the global minimum of free energy, are termed metastable structures. For instance, a large polymer crystal consisting of many extended chains is considered the most stable crystal. However, flexible polymer chains are usually folded, yielding thinner and only metastable lamellar crystals. The number of adjacent reentry chain folding in lamellar crystals of some polymers has been estimated via neutron scattering [68,69] and more recently via solid-state NMR [70,71] and via single-molecule force spectroscopy [72].

For a nucleus formed by one single chain, i.e., an intra-chain nucleus, this chain must be folded; otherwise, the interfacial energy would be too large [73]. For nuclei of small size, both for primary and secondary nuclei, the folded chain configuration is energetically

preferred. However, for nuclei consisting of many chains, the determination of the free energy is more complicated. An interplay of thermodynamic parameters and kinetic factors decides if a nucleus contains only stems from a single chain or if several chains participate in the formation of the nucleus. Factors, which govern adjacent reentry of folded chains are still not fully identified. Probably dependent on crystallization temperature, the formation of a nucleus is based on a compromise of intra-chain and inter-chain contributions. While intra-chain contributions may be thermodynamically preferred due to lower fold surface energy, inter-chain contributions may be kinetically favored due to a lower conformational entropy barrier [74].

Which measures allow to quantify the degree of metastability? The temperature at which the initially formed crystals melt (i.e., their "initial" melting point) and its deviation from the equilibrium melting point can be one important indicator of metastability. However, the determination of this "initial" melting point may be difficult or impossible as metastable crystals are prone to change in the course of (isothermal) crystallization or when heating these crystals. Nonetheless, fast DSC with high heating rates can be utilized and the temperature extrapolated to zero heating rate and zero crystallization time may yield values close to the "initial" melting point of the metastable crystals [75,76]. Recent experiments suggested that the annealing peak of semicrystalline polymers corresponds to melting of the originally formed metastable crystals, which due to constraints, did not yet have a chance to change (reduce their degree of metastability) [77]. In addition, parameters like lamellar thickness or the rarely considered width of crystalline structures may serve as suitable parameters for quantification of metastability [61].

Several reports have shown that the line of the Hoffman–Weeks plot ($T_M - T_C$ curve) is (over a certain temperature range) parallel to the $T_M = T_C$ line [61,78–82]. Having parallel lines (i.e., $T_M = T_C$+constant) does not allow extrapolation of Hoffman–Weeks plot for obtaining the equilibrium melting temperature. Observing parallel lines for crystals formed at low levels of crystallinity may be interpreted as the initially formed crystals did not have a chance (did not have sufficient time) to change and to reduce their degree of metastability beyond a certain (constant) amount. In particular, for low supercooling $T_M = T_C +$ constant is surprising because, at the corresponding low crystal growth rates, we expect that the available time is sufficient for lamellar thickening and widening to occur. Does $T_M = T_C +$ constant implies that at low supercooling lamellar crystals consist of crystalline blocks with a definite width, which does not change further in time? At present, we do not know the answer.

A crystal consisting of many fully extended chains is considered the thermodynamically most stable phase of polymer crystal. Examples are long-chain *n*-alkanes [83,84]. Extended chain crystals were also obtained via solid-state polymerization of crystallized monomers (topochemistry), crystallization under high-pressure, under external shear or by tensile drawing. Recently, Ye et al. obtained rather thick crystals of poly(butylene succinate) with a crystallinity close to 100% via a "green method" of washing out urea from urea/ poly(butylene succinate) inclusion compounds [85,86]. The lamellar thickness was estimated to be larger than 40 nanometers. Interestingly, in the melt, urea and poly(butylene succinate) are immiscible. However, they can form inclusion compounds, which have a melting point higher than that of each component, indicating strong urea/polymer interactions in the crystalline inclusion compound.

## 9. Structural Changes during Nucleation

During the process of nucleation, some order parameters, such as segmental orientation and chain conformation, will fluctuate more or less rapidly in time [87]. Previous experiments and simulations have revealed that the variation of these parameters with time does not show the same profile with prolonged time [88,89]. For instance, during aging below the glass transition ($T_g$), primary nucleation was observed to follow different trends related to enthalpic relaxation, i.e., nucleation happened only when the density

of the glassy state had reached a certain (threshold) value [90]. In other words, primary nucleation from the glassy state occurred after the densification of the glass.

Are preordered clusters formed before crystallization in solution or in the melt? Strobl has proposed a multi-stage crystallization process that involves a mesophase, i.e., an intermediate state between the amorphous melt and the final crystal phase [58,59]. The existence of a mesophase can explain several experimental observations, but this assumption was also highly debated. In challenging classical nucleation theories in the field of crystallization of small molecules or proteins, nucleation models have been proposed, which involve a mesophase [91–93]. However, it should be noted that for small molecules or proteins, two-step primary nucleation is mostly observed in solutions, where phase separation may first lead to the formation of an amorphous solid before crystal nucleation occurs.

In some studies, conformational ordering has been inferred before the onset of polymer crystallization [94–96]. Yan et al. revealed that on a highly oriented polyethylene (PE) substrate, poly(ε-caprolactone) (PCL) chains in the melt gradually became aligned parallel to the PE chains, which was attributed to a soft epitaxy effect [97]. Using micro-focus IR, Li et al. found preordering of polymer chains over a distance of around tens of microns away from the growth front of a spherulite [98]. Nonetheless, it cannot be ignored that there is an effect of the substrate and pressure gradients at the spherulite growth front on the process of ordering polymer chains. Conversely, in a real-time FTIR study of the isothermal crystallization of iPP and iPP random copolymers, Alamo and coworkers described the early-stages of primary nucleation led by density fluctuations while conformational preordering was clearly not observed prior to the onset of iPP crystallization [99]. Other recent synchrotron work led to the same conclusion [100]. Furthermore, more detailed molecular information of the ordering process is required for a deeper understanding of the early-stages of polymer crystallization.

After erasing all thermal history, the slow crystallization from the melt of random copolymers suggests that no preordered structures with enriched crystalline monomer units formed in the melt before crystal nucleation [62,99,101,102]. If such clusters had formed, one would have to expect that the spherulitic radial growth rate of the random copolymers should be comparable to that of the corresponding homopolymer. Thus, the ordering process responsible for secondary nucleation should take place directly at the growth front.

## 10. Interesting Observations and its Implications

Even at temperatures higher than its (equilibrium) melting point ($T_M^o$) PE [103] and PCL [104] still form adsorbed layers of ordered chains on graphite. These layers melt abruptly only at a temperature higher than $T_M^o$ . Upon cooling to the melting point of the bulk polymer, the transition to an ordered phase at the substrate interface, termed prefreezing, induced epitaxial crystallization and led to a logarithmic increase of the thickness of a crystalline layer. This process may account for the nucleating effect of graphite for PE and PCL. To quantitatively explain the variation of the pre-frozen layer thickness with superheating, a model considering the different interfacial free energies and the variation of interfacial potential with melt-substrate distance was proposed [103,105]. When epitaxially crystallized on a substrate of oriented PE, PCL shows lamellar crystals with a much higher lamellar thickness (55 nm) at the interface, which can propagate from the PE/PCL interface into the PCL bulk over a distance up to hundreds of nanometers [106]. What is the reason? Is it due to the high mobility of polymer chains in the crystalline layer at the interface between the two polymers, or has the presence of the substrate caused considerable changes in molecular interactions and thus conformations (reducing the entropic barrier)?

Kumaki et al. observed crystallization of PMMA chains within a compressed Langmuir–Blodgett film deposited on mica [107]. This was the first real-time observation of the crystallization of PMMA chains at a molecular level. Stepwise growth of folded chain blocks shorter than the chain was revealed, which differed from that expected from the classical Hoffman–Lauritzen theory. The authors concluded that "this observation indicated that

chains in the amorphous region might form a somewhat ordered structure before they attach to the crystal" [108]. The details of forming an ordered block-like structure are not clear yet. We speculate that it might result from intrachain nucleation. The stepwise growth behavior implies that multiple nucleation events are required to form a lamellar crystal. It is of particular interest that these blocks did not pack in a straight line. Blocks were separated frequently by steps, but these interconnected blocks at the crystal front shared the same chain direction. Does it mean that blocks were formed individually and adjusted their orientation upon attaching to the growth front of a lamellar crystal? Or are the blocks formed exactly at the growth front, and the steps separating the bocks are the result of thermal fluctuation reflecting a relatively low nucleation barrier? In the former case, the blocks should be formed due to primary nucleation, while in the latter, they are due to secondary nucleation without considering the influence of the substrate. Though the authors did observe an epitaxy effect of the mica substrate on the crystallization of PMMA chains, the chain directions were found to vary from crystal-to-crystal. Thus, a possible influence of the mica substrate on chain orientation in polymer crystals can be excluded.

## 11. Effect of Nucleation on the Semicrystalline Structure and Final Properties of Polymers

Nucleation, together with the interplay with chain dynamics, plays a very important role in determining semicrystalline structures and thus the final properties of polymer materials [19]. First, as aforementioned, the minimum lamellar thickness and the number of adjacent reentry chain foldings are determined by secondary nucleation. Depending on whether lamellar thickening happens or not during the crystallization process, the final lamellar thickness would deviate more or less from the minimum value. Lamellar thickness affects the melting point, which gives the upper limit of the application temperature window for crystalline thermoplastics. In addition, different crystal sizes will lead to distinct mechanical properties. For instance, Wang et al. obtained poly(lactide) film with nanocrystals via cold crystallization from a stretched melt, which demonstrated superior ductility and heat resistance compared to the standard material with large spherulites [109]. In fact, uniaxial and biaxial stretching has long been applied to produce semicrystalline plastic films yielding improved tensile strength, modulus, transparency and heat resistance. Second, for polymers with crystalline polymorphism, the kinetics of primary nucleation determines which crystal modification will prevail [19,110–116]. It has been reported that the solid–solid transformation from one to another crystal modification is usually initiated by nucleation [117,118]. Different polymorphs possess different properties, e. g., piezo- and pyroelectric β-polyvinylidene fluoride can be obtained from melt-recrystallization at atmospheric pressure through evaporation (in a vacuum) of a thin carbon layer on the surface of an ultrathin film of highly oriented α-polyvinylidene fluoride [119]. Third, the control of primary nucleation and crystallization of polymers has been applied to produce functional materials. For instance, Li et al. have built a platform using single polymer crystals as templates to produce functional nanomaterials [120,121]. Via increasing the number of interconnected edge-on poly(3-hexylthiophene) crystals, the lateral charge transport property could be improved [122]. Via controlling the primary nucleation density in a supersaturated solution, large needle-like poly(3-hexylthiophene), single crystals were obtained. These crystals revealed much stronger absorbance at long wavelengths than observed for drop-casted films and in the melt of this polymer [123].

To summarize, we show that crystal nucleation of polymer chains is far from equilibrium and that both thermodynamics and kinetics should be considered. In this viewpoint, we have summarized some open questions in the field and the recently proposed concepts. The starting melt may be non-equilibrated, leading to the observed melt memory effect. The molecular process and the pathways of chain folding during nucleation (both primary and secondary) are determined by both thermodynamics (free energy change) and kinetics (probability to choose a certain path). The chosen path and kinetics determine the originally formed lamellar thickness and crystallization rate, respectively. The hierarchical and cooperative ordering processes during nucleation have been revealed via both molecular

simulation and experimental observations; however, theories unifying these processes are still lacking. Via self-induced nucleation, the information of crystal lattice and orientation can propagate from the basal lamellar single crystal to other layers, leading to a stack of lamellar crystals in the registry. Via preliminary analysis of microscopic kinetics, we revealed that molecular interactions in crystals vary with nuclei size and that chain connectivity produces another entropy barrier for crystal nucleation of flexible polymer chains. We expect that the remaining open questions on nucleation can be solved by linking in a comprehensive approach: new theories treating kinetics in the non-equilibrium system, novel observations of molecular processes and simulations considering the peculiar molecular interaction in crystals (which is different from that in the melt). Answers to the above fundamental questions will not only be beneficial in the field of polymers but also provide profound insight for many basic problems in the growing field of non-equilibrium phenomena discussed in physics and chemistry.

**Funding:** J.X. thanks the National Natural Science Foundation of China (Grant nos. 21861132018 and 21873054) for financial support. G.R. acknowledges funding by the Deutsche Forschungsgemeinschaft (RE2273(18-1)). RGA acknowledges funding by the USA National Science Foundation, Polymer Program DMR 1607786.

**Data Availability Statement:** Data are contained within the article.

**Acknowledgments:** 

**Conflicts of Interest:** The authors declare no conflict of interest.

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
