# Peer review of "Concepts of Nucleation in Polymer Crystallization"

_crystals, doi:10.3390/cryst11030304_

Round 1

Reviewer 1 Report

The present paper is a review of recent works on polymer crystal nucleation.  The problem has many unsolved problems waiting future researches.  Many important concepts and related works being introduced, I think the present manuscript is a very good review and serves as an important source of information about the polymer nucleation problems.

However I want to point out, in the following, some issues I found hard to understand, the revisions of which will make the manuscript easier to read through.

  1. In the section 5 of secondary nucleation, the discussions are mainly on the LH-model and the authors’ one. However the appearance of equation (2) for the thickness of the 2D nucleus is rather abrupt without sufficient explanations.

  1. Misprint is on page 5 (“7. Origin of interfacial free energy”). Do the authors want to make a new section about it? In addition, the discussions on this and the next pages are not easy to understand.  Especially the word “the long range correlation number i*” in the caption for Fig.7 must be explained.  Greatest problem will be that no related papers are cited, which makes readers hard to fully understand its significance.

  1. The same is true also for section 7. The discussions seem to be insightful, but we cannot get deeper understanding without references being cited.

  1. General impression throughout the manuscript is that the primary nucleation, usually in 3D, and the secondary 2D nucleation are sometimes juxtaposed in the discussions, and the readers sometimes find it difficult to find which aspect of nucleation the authors are discussing. This problem is especially true when the authors referee to simulation studies, though no explanations about simulations are virtually made in this paper.        

Reviewer 2 Report

The review article provides a broad understanding on the Concepts of nucleation in polymer crystallization. This is a topic that people care about and hasn't been reviewed yet, though crystallization of polymer topic has been intensively studied in the past decades and even one of the authors has already write a review on Polymer Crystallization Theories. It is a well-written review where the key concepts and terminologies have been approached. As review article contains a large amount of detailed information.

The review is concise and focus in research conducted in the past decades but also on research from the past few years (2017: 8 references, 2018: 6, 2019:13 and 2020: 9).

A few typo corrections should be addressed:

Reference 9, line 422

Reference 31, line 463
